# 2nd Place Scheme on Action Recognition Track of ECCV 2020 VIPriors Challenges: An Efficient Optical Flow Stream Guided Framework

Haoyu Chen[1], Zitong Yu[1], Xin Liu[1], Wei Peng[1], Yoon Lee[2], and Guoying Zhao[1]

[1] CMVS, University of Oulu, Finland.
[2] CEL, Delft university of technology, the Netherlands.
{chen.haoyu, zitong.yu, xin,liu, wei.peng, guoying.zhao}@oulu.fi,
{y.lee}@tudelft.nl

**Abstract.** To address the problem of training on small datasets for action recognition tasks, most prior works are either based on a large number of training samples or require pre-trained models transferred from other large datasets to tackle overfitting problems. However, it limits the research within organizations that have strong computational abilities. In this work, we try to propose a data-efficient framework that can train the model from scratch on small datasets while achieving promising results. Specifically, by introducing a 3D central difference convolution operation, we proposed a novel C3D neural network-based two-stream (Rank Pooling RGB and Optical Flow) framework for the task. The method is validated on the action recognition track of the ECCV 2020 VIPriors challenges and got the 2nd place (88.31%) [1]. It is proved that our method can achieve a promising result even without a pre-trained model on large scale datasets. The code will be released soon.

**Keywords:** from-scratch training, 3D difference convolution, Rank Pooling, over-fitting

## 1 Introduction

Nowadays, with the strong ability of deep learning methods, training on massive datasets could consistently gain substantial performances on the action recognition task. However, it only works for a few very large companies that have thousands of expensive hardware GPUs and the majority of smaller companies and universities with few hardware clusters cannot enjoy the benefits. In this work, we try to train a model from scratch without large datasets or large scale pre-trained models while it can achieve state-of-the-art performance on the action recognition task.

Specifically, we introduce an enhanced convolution operation: 3D temporal central difference convolution (TCDC) into a traditional 3D CNN structure to

---

[1] https://competitions.codalab.org/competitions/23706#results

efficient spatio-temporal features in basic convolution operators with less over-fitting. Besides, instead of using raw RGB frames that might learn too much unnecessary details, we propose to use an efficient representation called Rank Pooling to serve as an enhanced RGB stream. Furthermore, the Optical Flow stream is used to guide the learning of the Rank Pooling stream to tackle the overfitting issue. At last, the Optical Flow stream and Rank Pooling stream are combined to be trained jointly on the task for better performance. The framework of our method is illustrated in Fig. 1. Our contribution to tackling this training-from-scratch task includes: a novel temporal convolution operator (3D TCDC), an Optical Flow guided Rank Pooling stream and a joint two-stream learning strategy for action recognition.

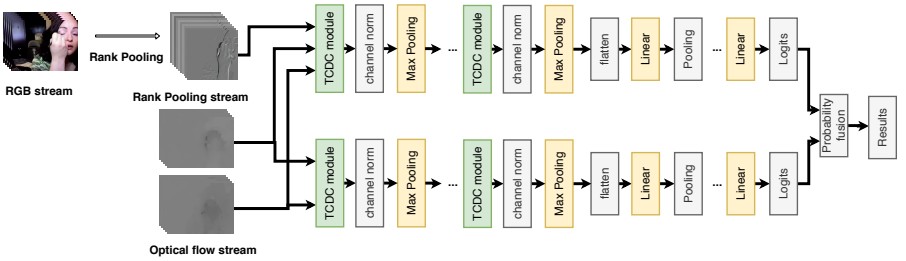

**Fig. 1.** Network architecture for our hybrid two stream framework. The Optical Flow is used to enhance the learning of Rank Pooling for overcoming the overfitting problem

## 2   Related work

The first common used two-stream 2D CNN architecture for action recognition was proposed by Simonyan and Zisserman [6], including one stream of RGB frames, and the other of Optical Flow. The two streams are trained separately and fused by averaging the scores of both the streams. A transition from 2D CNNs to 3D CNNs was made since the better performances of spatio-temporal features compared by 3D CNN to their 2D equivalents [8]. This transition comes with the problem of overfitting caused by small datasets and a high large number of parameters that need to be optimized [9] [10] in the model.

Specifically, in a two-stream (RGB and Optical Flow) framework, directly training models to learn RGB frames from scratch on a small dataset can lead to severe overfitting problem for RGB stream, while Optical Flow stream can still achieve relative high performances. The reason is that RGB frames contain too many noisy details and a large model could learn some irrelevant features which lead to overfitting with local optima. Many previous works have reported this overfitting issue, for instance, training from scratch on single RGB stream, 3D Resnet50 model [2] can achieve 55.2% accuracy, Slowfast model [3] for 40.1%, and even with neural network searching [5], the accuracy can only reach 61%.

To deal with the problem of overfitting, Carreira and Zisserman [1] introduced the Kinetics dataset with the I3D network, which was large enough to let 3D CNNs be trained sufficiently. Using RGB and Flow streams pre-trained on Kinetics [4], I3D achieved the state of art on the UCF101 [7] datasets. However, when the large scale datasets and pre-trained models are not available, especially for those who are not able to access to powerful computing facilities, how to overcome the overfitting is still an unsolved problem. In this work, we proposed to introduce a new 3D CNN operator TCDC [12], which is inspired by the 2D-CDC[13], and use Rank Pooling RGB stream with Optical Flow guided strategy to tackle this issue, which can achieve a promising result with a low computational cost.

## 3    Methodology

### 3.1    C3D Backbones with Central Difference Convolution

Based on the traditional 3D CNN framework [8], we introduce an unified 3D convolution operator called 3D temporal central difference Convolution (3D TCDC) for better integrating local gradient information. In a TCDC operation, the sampled local receptive field cube $\mathcal{C}$ is consisted of two kinds of regions: 1) the region in the current moment $\mathcal{R}'$, and 2) the regions in the adjacent moments $\mathcal{R}''$. In the setting of a TCDC, the central difference term is only calculated from $\mathcal{R}''$. Thus the generalized TCDC can be formulated as:

$$y(p_0) = \underbrace{\sum_{p_n \in \mathcal{C}} w(p_n) \cdot x(p_0 + p_n)}_{\text{vanilla 3D convolution}} + \underbrace{\theta \cdot (-x(p_0) \cdot \sum_{p_n \in \mathcal{R}''} w(p_n))}_{\text{temporal CD term}}. \tag{1}$$

where $w$, $x$ and $p$ denote the kernel weights, input feature maps and weight positions respectively. The first term in the right side stands for a Vanilla 3D convolution, while the second term stands for 3D TCDC operation. Please note that $w(p_n)$ is shared between vanilla 3D convolution and temporal CD term, thus no extra parameters are added. The hyperparameter $\theta \in [0, 1]$ is the factor to combine the contribution of gradient-level (3D TCDC) and intensity-level (Vanilla 3D). As a result, our C3D framework combines vanilla 3D convolution with 3D TCDC and could provide more robust and diverse modeling performance.

### 3.2    Rank Pooling for Optical Flow guided learning

We introduce a more explicit representation Rank Pooling instead of raw RGB frames to avoid the overfitting problem on the RGB steam. The definition of the Rank Pooling is below. Let a RGB stream sequence with k frames be represented as $< I1, I2, ..., It, ..., Ik >$, where $I_t$ is the average of RGB features over the frames up to $t$-timestamp. The process of Rank Pooling is formulated as following

objective function:

$$arg\,min \frac{1}{2}\left\|\omega\right\|^2 + \delta \sum_{i>j}^{\xi_{ij}}, \qquad (2)$$

$$s.t.\omega^T \cdot (I_i - I_j) \geq 1 - \xi_{ij}, \xi_{ij} \geq 0$$

By optimizing Eq. 2, we map a sequence of K frames to a single vector $d$. In this paper, Rank Pooling is directly applied on the pixels of RGB frames and the dynamic image $d$ is of the same size as the input frames. After the Rank Pooling images being generated, we combine the Rank Pooling stream with Optical Flow stream as input into the above C3D networks, which can enhance the learning of Rank Pooling stream.

## 4   Experiments

We validate our method on action recognition track of the ECCV 2020 VIPriors challenges with part (split 1) of the well-known action recognition dataset UCF101 [7]. There are 9537 video clips for training and validating, and 3783 for testing.

### 4.1   Different backbones

**Table 1.** Comparison of different backbone networks

| Backbone | Stream | Training Acc | Testing Acc | Overfitting gap |
|---|---|---|---|---|
| Slowfast[3] | RGB | 84.1% | 40.1% | 44.1% |
| Slowfast [3] | Optical Flow | 75.2% | 56.4% | 18.8% |
| ResNet 3D 101 [2] | RGB | 82.8% | 48.8% | 34.0% |
| ResNet 3D 101 [2] | Optical Flow | 84.4% | 66.3% | 18.1% |
| ResNet 3D 50 [2] | RGB | 84.1% | 51.8% | 32.3% |
| ResNet 3D 50 [2] | Optical Flow | 86.1% | 67.6% | 18.5% |
| NAS [5] | RGB | 88.9% | 50.2% | 38.7% |
| C3D [8] | RGB | 88.3% | 51.9% | 36.4% |
| C3D [8] | Optical Flow | 84.2% | 68.1% | 16.1% |
| **TCDC (ours)** | **RGB** | **91.4%** | **55.8%** | **35.6%** |
| **TCDC (ours)** | **Optical Flow** | **85.4%** | **77.2%** | **8.2%** |

In the experiment, we compared our 3D temporal CDC stacked networks (TCDC network) with C3D[8], ResNet 3D 50[2], ResNet 3D 101[2], SlowFast network[3] and also searched neural networks[5]. It turns out that our network performs the best among these networks. As shown in Table 1, we can see that our TCDC network can relatively solve the overfitting problem. However, there is still room to improve the performance, especially for the RGB stream. Then we introduce the Rank Pooling representations.

## 4.2   Efficiency of Rank Pooling stream

**Table 2.** Comparison of different stream fusions

| Fusing streams | Accuracy | | |
|---|---|---|---|
| Theta in TCDC network | 0.2 | 0.5 | 0.7 |
| RGB | 52.6% | 53.1% | 55.8% |
| RGB (Optical Flow enhanced) | 52.8% | 54.2% | 58.9% |
| Rank Pooling (Optical Flow enhanced) | 69.7% | 71.2% | 78.5% |
| Rank Pooling (Optical Flow enhanced) + Optical Flow | - | - | 83.8% |
| Rank Pooling (Optical Flow enhanced) + Optical Flow (ensemble 12 &16 frame) | - | - | **88.3%** |

To further overcome the serve overfitting problem of networks on RGB stream, we concatenate Optical Flow stream along with the RGB stream to enhance the learning procedure. However, as shown in Table 2, the benefit it gains is limited. We assume it's caused by the irrelevant features with local optima. Thus we propose to use a more explicit and efficient representation of RGB frames called Rank Pooling to tackle the problem. By introducing Rank Pooling representation, the overfitting problem is released (Rank Pooling 78.5% V.S. RGB 58.9%) as shown in third line of the Table 2. The best result is achieved by assembling the two stream results at clip lengths of 12 frame and 16 frame (all the data augmentations are implemented in all these frameworks).

## 4.3   Other experimental settings

Data augmentation techniques such as random cropping and horizontal flipping are proved very effective to avoid the problem of over-fitting. Here, we implemented two data augmentation techniques as same as [11]: 1. a corner cropping strategy, which means only 4 corners and 1 center of the images are cropped; 2. Horizontal Flip strategy that the training set is enlarged two times as the original one. We fix the input image size is 112*112. The clip length is 16 (ensembled with 12) frame. The optimal training parameters are set as 32, 0.1, 0.9, 10, 200 for the batch size, the initial learning rate, the momentum, the learning rate patience, and the epoch iteration respectively. The optimizer is standard SGD. The optical flow is extracted by a OpenCV wrapper for tvl1 optical flow and then processed by FlowNet2 [2] to generate 2-channel frames. The distribution platform is Pytorch with a single GPU: NVidia V100 (RAM: 32 GB).

---

[2] https://github.com/lmb-freiburg/flownet2-docker

## 5   Conclusions

In this work, we propose a data-efficient two-stream framework that can train the model from scratch on small datasets while achieving state-of-the-art results. By introducing a TCDC network on an Optical Flow guided Rank Pooling stream, we can substantially reduce the overfitting problem when dealing with small datasets. The method is validated on the action recognition track of the ECCV 2020 VIPriors challenges. It is proved that our method can achieve a promising result even without a pre-trained model on a large scale dataset.

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
