# OpenReview forum: "2nd Place Scheme on Action Recognition Track of ECCV 2020 VIPriors Challenges: An Efficient Optical Flow Stream Guided Framework"
_thecvf.com/ECCV/2020/Workshop/VIPriors — Submitted to VIPriors_

### Official Review · AnonReviewer2 · 2020-07-22
**Interesting method, unclear explanation**

**Confidence:** 4
**Rating:** 6

**Review:**

#### 1. [Summary] In 2-3 sentences, describe the key ideas, experiments, and their significance.
The paper proposes a new temporal convolution operator (3D TCDC), that combines a vanilla 3D convolution operator with a "Temporal Central Difference" term, and an optical flow guided Rank Pooling operation to compress the raw RGB input stream to a more compact single vector. A two stream network leverages both optical flow and the rank pooling representation to perform the task of action recognition. The method is evaluated on the VIPriors action recognition dataset and significant performance improvements are shown.

#### 2. [Strengths] What are the strengths of the paper? Clearly explain why these aspects of the paper are valuable.
* I like that the authors propose 3D TCDC as a new fundamental building block for 3D CNNs; the method seems interesting.
* The method seems effective on the dataset it has been evaluated on.

#### 3. [Weaknesses] What are the weaknesses of the paper? Clearly explain why these aspects of the paper are weak.
* The paper is too compact and omits some prerequisites that would make the method much more easy to understand. Especially section 3 would benefit from a more detailed explanation:
 * What does the temporal CD term in equation (1) do and what is the motivation behind it? It would have helped to provide a short recap of [13].
 * Similarly for the Rank Pooling operation: what is the motivation behind equation (2)? Which variable is optimized and what does the optimization represent? What are $\omega$, $\delta$, $\xi$? The authors could have at least referred to previous work on rank pooling.
 * How are the two streams fused, i.e. what is "Probability fusion" in Fig. 1?
* It would have been nice if the method would have been evaluated on multiple datasets.

#### 4. [Overall rating] Paper rating
* 6. Marginally above acceptance threshold

#### 5. [Justification of rating] Please explain how the strengths and weaknesses aforementioned were weighed in for the rating.
The proposed method seems interesting and effective. However, the paper provides too little explanation on the method. Nevertheless, I'm willing to accept the paper in the hope that the authors can further elaborate on the for the camera-ready version.

#### 6. [Detailed comments] Additional comments regarding the paper (e.g. typos or other possible improvements you would like to see for the camera-ready version of the paper, if any.)
* The abbreviation "C3D" is never explained.
* (line 202) "We assume it’s caused by the irrelevant features with local optima."
 Unclear explanation.
* See 3. for additional comments.

---

### Official Review · AnonReviewer1 · 2020-07-28
**2nd Place Scheme on Action Recognition Track of ECCV 2020 VIPriors Challenges: An Efficient Optical Flow Stream Guided Framework**

**Confidence:** 4
**Rating:** 4

**Review:**

1. [Summary] In 2-3 sentences, describe the key ideas, experiments, and their significance.

 This paper introduces a data-efficient pipeline to address the problem of action recognition. It is based on a two-stream model that utilizes an enhanced C3D network. The convolutions in the C3D are modified to include a 3D Temporal Central Difference Convolution term. Instead of working with RGB, authors proposed to use Rank Pooling guided with Optical Flow. Additionally, this work is ranked 2nd in de VIPriors Action Recognition Challenge.

2. [Strengths] What are the strengths of the paper? Clearly explain why these aspects of the paper are valuable.

 -	Modification of the convolution, integrating a new term.
 -	2nd Position in the challenge.

3. [Weaknesses] What are the weaknesses of the paper? Clearly explain why these aspects of the paper are weak.

 -	The paper feels more like a technical report than a proper paper.
 -	The introduction needs more motivation.
 -	Experiments are very much focused on the challenge. Only modifying the convolution would need more justification. A deeper study.


4. [Overall rating] Paper rating.

 4

5. [Justification of rating] Please explain how the strengths and weaknesses aforementioned were weighed in for the rating.

 Weaknesses of point 3 justify the rating.

6. [Detailed comments] Additional comments regarding the paper (e.g. typos or other possible improvements you would like to see for the camera-ready version of the paper, if any.)

 Please, include Rank Pooling citation, it seems is introduced by the authors.

---

### Decision · Program_Chairs · 2020-07-29

**Decision:**

Reject

**Comment:**

After considering the reviews and further discussion we concluded that although the idea seems interesting, the paper was hard to read and the method was not clearly explained. We therefore believe that the paper needs one more iteration before it could be accepted.